# Use of Clopidogrel and Proton Pump Inhibitors Alone or in Combinations in Persons with Diabetes in Denmark; Potential for CYP2C19 Genotype-Guided Drug Therapy

**DOI:** 10.3390/metabo11020096

**Published:** 2021-02-10

**Authors:** Niels Westergaard, Lise Tarnow, Charlotte Vermehren

**Affiliations:** 1Centre for Engineering and Science, Department of Biomedical Laboratory Science, University College Absalon, Parkvej 190, 4700 Naestved, Denmark; 2Steno Diabetes Center, Birkevaenget 3, 3rd, 4300 Holbaek, Denmark; litar@regionsjaelland.dk; 3Department of Clinical Pharmacology, Bispebjerg Hospital, University of Copenhagen, Bispebjergbakke 23, 2400 Copenhagen, Denmark; charlotte.vermehren@regionh.dk; 4Department of Pharmacy, Section for Social and Clinical Pharmacy, Faculty of Health and Medical Sciences University of Copenhagen, Universitetsparken 2, 2100 Copenhagen, Denmark

**Keywords:** clopidogrel, proton pump inhibitors (PPIs), drug use, pharmacogenomics, polypharmacy, drug–drug interactions, drug–gene interactions, drug–drug–gene interactions, diabetics, elderly, cardiovascular disease

## Abstract

Background: Clopidogrel and proton pump inhibitors (PPIs) are among the most used drugs in Denmark for which there exists pharmacogenomics (PGx)-based dosing guidelines and FDA annotations. In this study, we further scrutinized the use of clopidogrel and PPIs when prescriptions were redeemed from Danish Pharmacies alone or in combination in the Danish population and among persons with diabetes in Denmark. The focus deals with the potential of applying PGx-guided antiplatelet therapy taking both drug–drug interactions (DDI) and drug–gene interactions (DGI) into account. Methods: The Danish Register of Medicinal Product Statistics was the source to retrieve consumption data. Results: The consumption of PPIs and clopidogrel in terms of prevalence (users/1000 inhabitants) increased over a five-year period by 6.3% to 103.1 (PPIs) and by 41.7% to 22.1 (clopidogrel), respectively. The prevalence of the use of clopidogrel and PPIs in persons with diabetes are 3.8 and 2.1–2.8 times higher compared to the general population. When redeemed in combination, the prevalence increased to 4.7. The most used combination was clopidogrel and pantoprazole. Conclusions: The use of clopidogrel and PPIs either alone or in combination is quite widespread, in particular among the elderly and persons with diabetes. This further supports the emerging need of accessing and accounting for not only DDI but also for applying PGx-guided drug therapy in clinical decision making for antiplatelet therapy with clopidogrel having a particular focus on persons with diabetes and the elderly.

## 1. Introduction

The use of drugs in Denmark having pharmacogenomics (PGx)-based actionable dosing guidelines (AG) from either CPIC and/or DPWG for CYP2D6 and CYP2C19 drug–gene pairs were measured in two recent studies [1,2]. It was shown that a large fraction of the Danish population, especially the elderly living in nursing homes, are exposed to drugs or drug combinations for which AG exists. Interestingly, clopidogrel as well as proton pump inhibitors (PPIs) turned out to be among the most used drugs having AG both in the general population and among nursing home residents. In this study, we will further scrutinize the use of clopidogrel and PPIs when prescriptions of these drugs were redeemed from Danish pharmacies alone or in combination in the general population and among persons with diabetes in Denmark. The study focuses on drug–drug interactions (DDI), drug–gene interactions (DGI) and the potential of applying PGx-guided antiplatelet therapy.

Clopidogrel, prasugrel and ticagrelor, all P2Y12 platelet inhibitors [3,4], are the standard-of-care oral drugs used in thrombosis prophylaxis. This treatment is widely used among the elderly and in persons with diabetes having a well-known high risk for cardiovascular diseases and increased platelet reactivity, i.e., secondary prophylaxis in cerebral infarction and acute coronary syndrome, respectively [3,5,6,7]. The treatments have been shown to inhibit blood clots in coronary artery disease, peripheral vascular disease and cerebrovascular disease, and to prevent myocardial infarction [3,8,9]. Although clinical practice guidelines now recommend prasugrel and ticagrelor over clopidogrel, in acute coronary syndrome patients undergoing percutaneous coronary interventions (PCI), clopidogrel remains the most widely used P2Y12 inhibitor [3,9]. Clopidogrel is a prodrug that requires cytochrome P450 biotransformation, primarily by CYP2C19, to form an active metabolite that selectively and irreversibly inhibits the purinergic P2Y12 receptor and, thus, platelet aggregation for the life span of the platelet [3,9].

In this study, we focus on clopidogrel as the object drug, i.e., changes in pharmacokinetic and/or pharmacodynamic properties of clopidogrel caused by either genetic features impacting CYP2C19 metabolic activity and/or by concomitant intake of precipitant drugs, in this case PPIs [10,11,12]. Genetic polymorphisms of CYP2C19 are associated with impaired metabolism of clopidogrel [8,10], referred to as DGI, leading to higher cardiovascular event rates compared to normal function [8,13,14]. PGx-based AG have been issued from CPIC [10], DPWG [15] and RNPGx [16] all dealing with the CYP2C19 phenotypes (genotypes); intermediate—(IM) and poor metabolizers (PM) (see also www.pharmgkb.org (accessed on 10 February 2021)). In addition, the FDA-approved drug label for clopidogrel includes PGx annotations for PM obviating consideration for alternative drugs [8,17] independently of diagnosis. The American Stroke Association secondary stroke guideline [18], and the Report of the American College of Cardiology Foundation Task Force on Clinical Expert Consensus Documents, and the American Heart Association [19] note that PM can have reduced platelet inhibition. PGx testing for CYP2C19 may be considered before starting clopidogrel therapy in PCI patients believed to be at moderate or high risk for poor outcomes.

Clopidogrel is associated with increasing risks of gastrointestinal (GI) bleeding both when given alone or in combination with aspirin [4,20]. Consequently, clopidogrel is commonly prescribed in combinations with PPIs to prevent GI bleeding [4,7,20,21]. According to Flockhart and others, the PPIs are inhibitors of CYP2C19 with esomeprazole reported as a strong inhibitor and pantoprazole, lansoprazole and omeprazole as weaker inhibitors [12,21,22]. Besides being inhibitors of CYP2C19, PPIs are also primarily metabolized by CYP2C19 and PGx-based AG dealing with rapid metabolizers (RM) for pantoprazole, lansoprazole and omeprazole but not for esomeprazole [11,23]. The FDA and the EMA published safety concerns of concomitant use of clopidogrel and PPIs in 2009 and 2010 (see [24,25,26]). Inconsistent effects on cardiovascular outcomes of concomitant use of clopidogrel and PPIs have been reported [27,28]. However, clinically important interactions cannot be excluded as substantiated in several literature studies and meta-analysis showing that the combined use of clopidogrel and PPIs are associated with increased adverse cardiovascular events, such as major adverse cardiac events (MACE), stent thrombosis (ST) and myocardial infarction (MI) following PCI [20,24,29,30]. The concomitant use of clopidogrel and PPIs exhibit the Achilles’ heel of personalized medicine, referred to as phenoconversion, which describes the overlapping of DDI and DGI [31,32], also denoted to as drug–drug–gene interaction (DDGI). In spite of a strong association between CYP2C19 gene variants and outcomes related to antiplatelet therapy [9,33], only a few studies have actually dealt with DDGI in relation to concomitant use of PPIs [34,35]. Altogether, this warrants the need for special caution when it comes to the use of clopidogrel alone or in combination with PPIs to balance overall risk and benefits. The purpose of this study is to underpin the potential of applying PGx tests enlightened by user data, with particular focus on persons with diabetes, as a supportive tool prior to initiating treatment with clopidogrel alone or in combination with PPIs.

## 2. Results

Table 1 shows the consumption of PPIs and platelet aggregation inhibitors from 2014 to 2018 in Denmark. As can be seen, the use of PPIs both in terms of number of users and prevalence increased over time with a total increase of 9.2% in terms of users and 6.3% in terms of prevalence. The consumption of PPIs includes esomeprazole, lansoprazole, omeprazole and pantoprazole and rabeprazole; however, the use of rabeprazole was less than 255 users per year. The same pattern was seen for clopidogrel. The use increased yearly with a total increase of 45.6% in terms of users and 41.7% in terms of prevalence. The use of prasugrel and ticagrelor was very low compared to clopidogrel, and no increase in use was seen.

Figure 1 shows the age-distribution expressed as prevalence (users /1000 inhabitants) (A) and total number of users (B) who redeemed prescriptions of clopidogrel and proton pump inhibitors during 2018. As can be seen, the prevalence of use increased with age, in particular for clopidogrel and pantoprazole. The total number of users for each drug are given in Table 2, showing that pantoprazole is the most widely used PPI followed by lansoprazole, omeprazole and esomeprazole. 

Table 2 shows the consumption in terms of number of users among the general population and persons with diabetes who redeemed prescriptions of clopidogrel and PPIs either alone or in combination. The table also shows the number of users who redeemed clopidogrel and PPIs on the same day or during 2018, respectively. The term “used or redeemed alone” does not exclude that the users could have redeemed prescriptions of other drug combinations. From the table it can be seen that, e.g., 127,480 persons redeemed prescriptions of clopidogrel, 329,222 pantoprazole, 25,641 the combination of clopidogrel and pantoprazole during 2018, and 13,850 redeemed the combination on the same day. For persons with diabetes the same numbers were 21,746, 39,287 and 5285, and on the same day 2876. The use of PPIs among persons with diabetes followed the same pattern as the general population, i.e., pantoprazole is the most used PPI followed by lansoprazole, omeprazole and esomeprazole both when redeemed alone or in combination with clopidogrel. It should be noted that the number of users who redeemed prescriptions of the combinations of clopidogrel and PPIs on the same day is a subset of the number for the whole year. Moreover, the number of users for the different PPIs are not additive, since dispensing to the same user can occur for the different PPIs. The number of users who redeemed the combinations of clopidogrel and PPIs on the same day as percentage of the whole year ranged from 51 to 54%. 

On the basis of the data in Table 2 and the sizes of the total Danish and diabetic populations (see Materials and Methods), the prevalences of use of clopidogrel and PPIs redeemed either alone or in combination were calculated and are presented in Table 3. The fraction of users who redeemed prescriptions of the combination of clopidogrel and PPIs relative to clopidogrel was significantly higher among persons with diabetes, both when measured on the same day and during 2018, suggesting that persons with diabetes are more exposed to the combination of clopidogrel and PPIs than the general population. In addition, the prevalence ratios, i.e., prevalence diabetics/ prevalence general population, are shown in Table 4. The prevalence of use of clopidogrel alone, among persons with diabetes, was almost four times higher compared to the general population, whereas the same numbers for PPIs ranged from 2.1 to 2.8. However, when clopidogrel and PPIs are redemed in combination, either on the same day or during 2018, the prevalence ratios increased to around 4.7, which is significantly higher as compared to that found for clopidogrel and the PPIs when redeemed alone.

Table 5 shows warnings related to potential DDIs for the combinations of clopidogrel and PPIs obtained from different providers of open access drug interaction trackers. The aim is to visualize the differences in warnings among the different trackers as well as give numbers of how many individuals, both in the general population and among persons with diabetes, potentially could be affected by co-administration of clopidogrel and PPIs. None of the trackers scored the combination of clopidogrel and pantoprazole as a “serious use alternate”. This combination was the most frequent combination observed both in the general population (25,641) and among persons with diabetes (5285). However, for the other combinations of clopidogrel and PPIs, discordance between the different trackers was seen for “serious use alternate” and “monitor closely”.

## 3. Discussion

The Danish Register of Medicinal Product Statistics [40], which comprises records of drug prescriptions redeemed to individuals, offers unique opportunities to study drug use, including combinations thereof as previously shown [1,2] without compromising users’ identity. This has made it possible to map the consumption of drugs having PGx-based AG as well as to measure the occurrence of inappropriate drug combinations at ATC level 7 (drug name level). On the basis of previous findings of the wide use of clopidogrel and PPIs [1,2], this study further scrutinized the use of these drugs in the general population and among persons with diabetes, who in particular have a high risk for cardiovascular events [5,6], in order to underpin the potential of applying PGx test in clinical decision making. It should be emphasized that this register study does not provide any information about diagnosis, dose, compliance, clinical effects or duration of treatments. 

The finding that the consumption of clopidogrel has increased quite significantly (45.6%) over a five-year period is in alignment with a shift in Danish recommendations for cerebral infarction prophylaxis, from a combination treatment consisting of acetyl salicylic acid and dipyridamole to clopidogrel as the first choice treatment [41].

The use of PPIs increases to a lesser extent over the same period. There is a general suspicion of a large over-consumption of inappropriately prescribed PPIs, which is why de-prescribing guidelines have been developed [42]. However, PPIs may be indicated when treated with drugs associated with ulcerogenic and gastrointestinal side effects such as clopidogrel and metformin. However, still, approximately 10% of the Danish population redeems PPIs every year, and importantly, the prevalence of the use of both clopidogrel and PPIs increased, as a function of age, to as high as 122 and 174 users/1000 inhabitants, respectively. Recent data from 2019 showed that the use of clopidogrel and PPIs continued to increase both in terms of users (133,430 and 610,755) and prevalence (23 and 105) [40].

Nationwide cohorts to determine drug consumption not only for single drug use, as discussed above, but also for use of drug combinations in patient subgroups have not, to our knowledge, been determined before at a level this detailed. In this study, we further scrutinized the use of clopidogrel and PPIs when prescriptions were redeemed alone or in combination on the same day or during 2018 in the general population and in persons with diabetes. Out of a population of 5.8 million inhabitants, 258,494 redeemed prescriptions of ATC code A10 (drugs used in diabetes) corresponding to 4.5% of the Danish population. This finding is compatible with a recent study exploring the prevalence of diabetes in the Danish population [43]. 

Persons with diabetes are more prone to cardiovascular diseases and increased platelet reactivity [3,5,6], which is reflected in the high prevalence of use of clopidogrel; this is almost four times higher compared to the general population. The prevalence of PPIs in persons with diabetes follows the same pattern, i.e., 2.1–2.8 times higher. Interestingly, when clopidogrel and PPIs were redeemed in combination either on the same day or during 2018, the prevalence ratios increased to around 4.6, which is significantly higher compared to that found for clopidogrel and the PPIs when redeemed alone. This could suggest that persons with diabetes using clopidogrel per default are being prescribed more PPIs than the general population. One possible explanation could also be the frequent treatment of type 2 diabetes with metformin, which has well-known gastrointestinal side effects and was reedemed by 181,485 users in 2018 [40]. In addition, it has been found that persons with type 2 diabetes have a significantly higher risk of developing peptic ulcers after adjustment for possible confounding factors [44].

Comparative studies have suggested to what extent PPIs reduce the conversion of clopidogrel to the active metabolite and thereby attenuating the antithrombotic effect of clopidogrel [22]. The strongest inhibitor was omeprazole, followed by esomeprazole, then lansoprazole, and no effect was reported for pantoprazole [12,21,22]. In this study, we used four open access “drug interaction checkers” to score for severity of DDI of the combinations of PPIs and clopidogrel. Three of four checkers scored the combinations of clopidogrel and omeprazole or esomeprazole as a “serious use alternate”, which is also in alignment with the FDA and EMA recommendations [21,24,25,26]. The most frequent combinations of clopidogrel and PPIs were for clopidogrel and pantoprazole followed by lansoprazole, omeprazole and esomeprazole. Direct comparison of prevalence data in this study for combination of clopidogrel and PPIs to other studies is not possible due to lack of similar cohort studies; however, data supporting the widespread use of combinations of clopidogrel and PPIs can be found [45,46]. In this context, it should be pointed out that persons with diabetes have a higher prevalence of concomitant use of clopidogrel and PPIs. One should bear in mind that the number of users co-administrating clopidogrel and PPIs refers to the Danish situation. In this situation actions should be taken, cf. the Danish interaction checker, which scores the co-administration of clopidogrel and PPIs as “monitor closely” for all PPIs, while the other checkers consider the combinations of clopidogrel and the various PPIs to be “serious use alternate”—particularly with regard to esomeprazole and omeprazole. The authors’ experience with this warning is that it is rarely taken into consideration due to its vague formulation, unless the user(s) undergo a thorough medication review. 

In spite of supporting evidence and advances in PGx implementation in clinical practice over the last decade, significant barriers still exist. Mainly concerning physicians’ and pharmacists´ awareness and education, but also evidence level, significance and cost-effectiveness are questioned [47]. In Denmark the situation does not look much different, and the use of PGx tests have not gained foothold in daily clinical practice [1,2,48]. Taking the average Caucasian frequencies of DGI recently reported [49] for CYP2C19 (IM 26.9% and PM 2.6%) into consideration further suggests that a significant proportion of the Danish population, and persons with diabetes using clopidogrel, will have DGI for which actions in principle should be taken regarding dose adjustments corresponding to 37,600 and 6415, respectively (calculation based on data from Table 2). If the impact of the phenotypes PM and IM are taken into consideration as well by “drug interaction checkers”, the balance between “monitor closely” and “serious use alternate” might change significantly towards the latter for combinations of PPIs, especially for omeprazole and esomeprazole, which have been corroborated recently [34,35]. This further substantiates that DGI and DDGI are unrecognized in clinical practice in general and in the elderly [1,2] and in persons with diabetes in particular, who are more exposed to polypharmacy [50]. These added risks are likely to result in increased health care costs [51]. Supportive evidence on cost-effectiveness of applying CYP450 PGx-guided antiplatelet therapy preemptively in cardiovascular diseases [52] and in polypharmacy have emerged [53]; however, still more studies are needed. In addition, a newly published study has shown that regular use of PPIs has been associated with a higher risk of type 2 diabetes, and the risk increased with longer duration of use. These findings add another dimension to exercise caution when prescribing PPIs, particularly for long-term use [54].

In conclusion, we have demonstrated that the use of clopidogrel and PPIs, both having PGx-based actionable dosing guidelines and FDA annotations, either given alone or in combination, is quite widespread, in particular among persons with diabetes and the elderly in Denmark. This further supports the emerging results of acccessing and accounting for not only DDI but also for DGI and DDGI as supportive tools in clinical decision making for antiplatelet therapy, which is, in addition, likely to be cost-effective.

## 4. Materials and Methods

The Danish Register of Medicinal Product Statistics [40] that comprises records of all prescriptions redeemed since 1 January, 1996, was used to retrieve drug consumption by using Medstat.dk [40] and by support of Statistics Denmark [55]. The personal identification number [56] (the CPR number) is a unique identifier to all Danish inhabitants that makes it possible to measure a person’s drug consumption. Consumption is expressed as the number of users who redeemed prescriptions of clopidogrel and/or PPIs alone or in combination either on the same day or during 2018 by applying their ATC codes [57]. The number of persons with diabetes is identified by measuring inhabitants who redeemed prescriptions of the ATC code A10 (level 2), which solely includes “drugs used in diabetes”, i.e., A10A (insulins and analogues) and A10B (blood glucose lowering drugs excl. insulins) during 2018. By combining A10 to the ATC codes for clopidogrel and the PPIs, the number of persons with diabetes who redeemed prescriptions of clopidogrel and PPIs alone or in combination either on the same day or during 2018 was identified. Over the counter (OTC) consumption is not identifiable with a person and therefore is not part of this study; however, it should be noted that PPIs in low strengths and small package sizes are available as OTC products, and this accounts for around 3% of the total consumption. 

To convert number of users to prevalence (users/1000 inhabitants), the total Danish population in 2018 was 5,781,190, and the age group distribution was as follows: 0–17 years 1,165,000; 18–24 years 532,622; 25–44 years 1,441,697; 45–64 1,525,308; 65–79 years 859,369 and 80+ 256,694. The total number of persons who redeemed prescriptions of ATC-code A10 (persons with diabetes) was 258,494, and the age group distribution was as follows: 0–17 years 3107; 18–24 years 3695; 25–44 years 23,685; 45–64 94,880; 65–79 years 103,926 and 80+ 29,201.

Drug–drug interactions were scored in severity by using “interaktionsdatabasen” managed by the Danish Medicine Agency [36]; Felleskatalogen [37] (Norway), Medscape^®^ drug interaction checker [38] and Drugs.com (accessed on 28 December 2020). [39]. Warnings related to CYP2C19 are displayed as “monitor closely” or “serious use alternate”. The dosing information, length of treatment and indication for prescribing were not recorded, and ethics approval was not applicable according to Danish law, since the use of anonymized healthcare data for pharmacoepidemiological research does not require subject consent or approval from the Ethics Committee.

## Figures and Tables

**Figure 1 metabolites-11-00096-f001:**
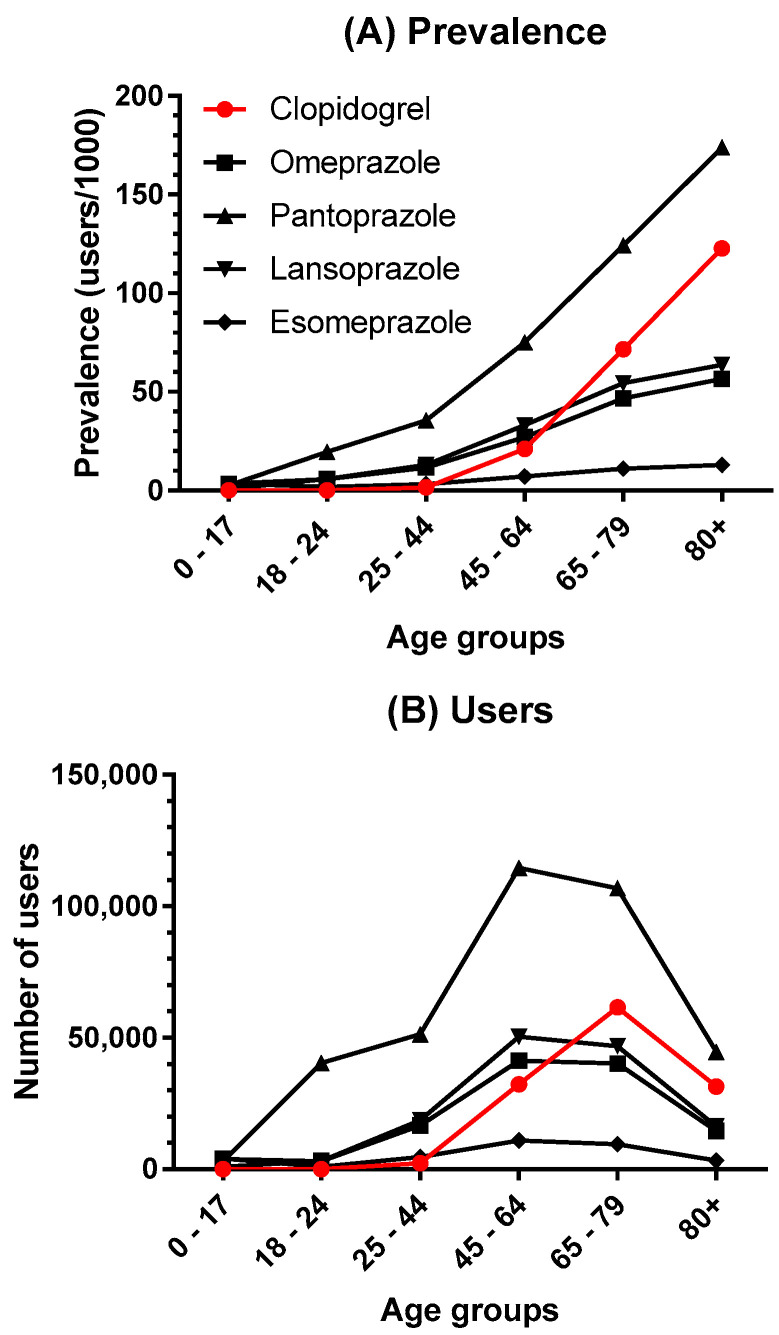
Prevalence (**A**) and total number of users (**B**) of clopidogrel and proton pump inhibitors in 2018 as a function of age groups. For legends see (**A**).

**Table 1 metabolites-11-00096-t001:** Consumption of proton pump inhibitors (PPIs) and platelet aggregation inhibitors in the Danish population during 2014–2018.

	2014	2015	2016	2017	2018
PPIs A02BC	545,990 (97.0)	570,745 (100.8)	583,345 (102.4)	591,195 (102.8)	596,035 (103.1)
Clopidogrel B01AC04	87,770 (15.6)	100,835 (17.8)	111,315 (19.5)	119,735 (20.8)	127,755 (22.1)
Prasugrel B01AC22	1460 (0.3)	1035 (0.2)	575 (0.1)	360 (<0.1)	325 (<0.1)
Ticagrelor B01AC24	9345 (1.7)	9605 (1.7)	9790 (1.7)	9465 (1.7)	9500 (1.7)

Data are presented as total number of users who redeemed prescriptions of PPIs and platelet aggregation inhibitors during the period of 2014–2018. Numbers in brackets are prevalence (number of users/1000 inhabitants). PPIs constituted of esomeprazole, lansoprazole, omeprazole, pantoprazole and rabeprazole.

**Table 2 metabolites-11-00096-t002:** Consumption in terms of number of users who redeemed prescriptions of clopidogrel and proton pump inhibitors among the general population and in persons with diabetes in Denmark in 2018.

		Esomeprazole A02BC05	Lansoprazole A02BC03	Omeprazole A02BC01	Pantoprazole A02BC02
Users:		**32,295**	**135,980**	**119,274**	**329,222**
Clopidogrel	**127,480**	2388 [1217]	9570 [5213]	7188 [3900]	25,641 [13,850]
Users with diabetes:		**3054**	**17,246**	**14,286**	**39,287**
Clopidogrel	**21,746**	484 [250]	1952 [1081]	1459 [813]	5285 [2876]

Data are presented as number of users and number of users with diabetes who redeemed clopidogrel (B01AC04) and proton pump inhibitors either alone (in bold) or in combination during 2018 or on the same day during 2018 (numbers in square brackets). The ATC codes for the different drugs are given.

**Table 3 metabolites-11-00096-t003:** Consumption of clopidogrel and proton pump inhibitors among the general population and in persons with diabetes in Denmark in terms of prevalence.

		Esomeprazole A02BC05	Lansoprazole A02BC03	Omeprazole A02BC01	Pantoprazole A02BC02
Prevalence:		**5.6**	**23.5**	**20.6**	**56.9**
Clopidogrel	**22.1**	0.4 (1.8%) [0.2 (0.9%)]	1.7 (7.7%) [0.9 (4.1%)]	1.2 (5.4%) [0.7 (3.2%)]	4.4 (20.0%) [2.4 (10.9%)]
Prevalence diabetics:		**11.8**	**66.7**	**55.3**	**152.0**
Clopidogrel	**84.1**	1.9 (2.3%) * [1.0 (1.2%)] *	7.6 (9.0%) * [4.2 (5.0%)] *	5.6 (6.7%) * [3.1 (3.7%)] *	20.4 (24.3%) * [11.1 (13.2%)] *

Data are presented as prevalence general population (number of users/1000 inhabitants) and prevalence diabetics (number of diabetic/1000 diabetics) who redeemed prescriptions of clopidogrel and PPIs either alone (in bold) or in combination during 2018 or on the same day during 2018 (numbers in square brackets). Numbers in brackets express the percentage of users who redeemed the combinations of clopidogrel and PPIs relative to clopidogrel. * *p* ≤ 0.05 or less compared to the corresponding values for prevalence in the general population (chi-squared test).

**Table 4 metabolites-11-00096-t004:** Consumption of clopidogrel in combination with proton pump inhibitors in terms of prevalence ratios.

Prevalence Ratio: (Prevalence Diabetes/Prevalence General Population)			Esomeprazole	Lansoprazole	Omeprazole	Pantoprazole
			**2.1**	**2.8**	**2.7**	**2.7**
Clopidogrel	**3.8**	During 2018	4.8	4.5	4.7	4.6
		Same Day 2018	5.0	4.6	4.4	4.6

Data are presented as prevalence ratio (prevalence diabetes/prevalence general population) based on the prevalence shown in Table 3.

**Table 5 metabolites-11-00096-t005:** Drug–drug interaction tracker scores for the combination of clopidogrel and PPIs.

		Esomeprazole	Lansoprazole	Omeprazole	Pantoprazole
clopidogrel	(1)				
	(2)				
	(3)				
	(4)				
Total number of users * Total number of diabetic users *		2388	9570	7188	25,641
		484	1952	1459	5285

Drug–drug interactions were scored by using four different open access databases (1) Interaktionsdatabasen [36] (2) Felleskatalogen [37] (3) Medscape [38] and (4) Drugs.com (accessed on 28 December 2020). [39]. Yellow: “monitor closely” and red “serious use alternate”. Warnings are related to CYP2C19 activity. * Total number of users in the general population and among diabetics who redeemed prescriptions of the combination of clopidogrel and PPIs during 2018 (data from Table 2).

## Data Availability

The data presented in this study are available within the article.

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
