# Peer review of "Use of Clopidogrel and Proton Pump Inhibitors Alone or in Combinations in Persons with Diabetes in Denmark; Potential for CYP2C19 Genotype-Guided Drug Therapy"

_metabolites, 2021, doi:10.3390/metabo11020096_

Round 1

Reviewer 1 Report

The manuscript follows an interesting idea but it seems to lack a clear/novel conclusion. Authors stress the increased use of the target drugs, but this cannot be a conclusion per se. 

So, I suggest the following:

-a change of the title with the aim of transmitting a clear message based on the results of the study. 

-a conclusion based on the prevalence study in correlation with the PGx and AG suggesting a clear measure that should be taken regarding the used of the target drugs, especially in DM /ageing patients

-including a conclusion in the abstract in order to give the reader a hint about the content of the manuscript

Author Response

Dear reviewer,

Thank you for your thorough review and our responses can be seen below. We fully agree with all suggestions proposed which can be seen in the revised manuscript in track changes mode.

  • -a change of the title with the aim of transmitting a clear message based on the results of the study. 
  • -a conclusion based on the prevalence study in correlation with the PGx and AG suggesting a clear measure that should be taken regarding the used of the target drugs, especially in DM /ageing patients
  • -including a conclusion in the abstract in order to give the reader a hint about the content of the manuscript

All suggestions have been incorporated into the manuscript as requested

Reviewer 2 Report

The work you have done using real-world data ( The Danish Register of Medicinal Product Statistics) is very interesting and the focus on potential drug-drug interactions, drug-gene interactions and the potential of applying pharmacogenomics based dosing guidelines for guided antiplatelet therapy very important. 

The conclusion should put more emphasis on the need to consider the use of pharmacogenomics based dosing guidelines

Knowing that there are drugs that are recommended by guidelines for reducing vascular risk and peptic ulcer, and also that there are differences determined by individual characteristics in the effectiveness of the same drugs either when prescribed alone or prescribed together, Knowing, also that some of de differences are genetically determined The authors conducted a study on a database of medicines supplied in pharmacies in which they analyzed the dispensing of PPI and Clopidogrel both in the general population and in diabetes. Within the PPI class, it is known that there are drug differences in the interactions and that these must be taken into account. They concluded that there is a significant, and increasing, use of Clopidogrel, either alone or in conjunction with PPI. Within this group, there's a significative use of molecules with a potential effectiveness-reducing interaction. The study warns of the need to take into account individual variations of a genetic basis and within a family of drugs to use drugs with less interference and having into consideration the need to consider pharmacogenetic guidelines.

In the abstract eventually, the paragraphs in lines 20-24, could be inverted
table 2 first line is difficult to read; eventually, text orientation could be changed (line 142)

I think the article is fundamental for two reasons.

1 - The use of information derived from national drug redeemed registries during several years
2 - The need to have in mind genetically determined interactions to maximize medication efficacy and reducing drug-related risks

Author Response

Dear reviewer,

Thank you for your thorough review and our responses can be seen below. We fully agree with all suggestions proposed which can be seen in the revised manuscript in the track changes mode.

  • The conclusion should put more emphasis on the need to consider the use of pharmacogenomics based dosing guidelines

Conclusion changed accordingly as this was also a point of reviewer 1

  • In the abstract eventually, the paragraphs in lines 20-24, could be inverted
    table 2 first line is difficult to read; eventually, text orientation could be changed (line 142)

Changed as suggested

Reviewer 3 Report

The manuscript entitled “Use of Clopidogrel and Proton Pump Inhibitors and Combinations thereof Nationwide and Among Persons with diabetes in Denmark; Call for Actions” describes that the consumption of PPIs and clopidogrel in terms of prevalence (users/1000 inhabitants) increased over a five-year period by 6.3 % to 103.1 (PPIs) and by 41.7 % to 22.1 (clopidogrel), respectively. The prevalence of use of clopidogrel and PPIs in persons with diabetes are 3.8 and 2.1-2.8 times higher, respectively, compared to the general population. When redeemed in combination this number increased to 4.7. The most used combination is clopidogrel and the PPI, pantoprazole. It demonstrates that the use of clopidogrel and PPIs either alone or in combination is quite widespread, in particular among persons with diabetes in Denmark. This further supports the emerging results of acccessing and accounting for not only DDI but also for DGI and DDGI as a supportive tool in clinical decision making for antiplatelet therapy.

English language and style check are required.

Author Response

Dear reviewer,

Thank you for your thorough review and our response can be seen below. We fully agree with the suggestion proposed which can be seen in the revised manuscript in the track changes mode.

  • English language and style check are required.

This request has been accomplished.

Round 2

Reviewer 1 Report

Title suggestion- Use of Clopidogrel and Proton Pump Inhibitors Alone or in Combinations In Patients with Diabetes in Denmark; Potential for CYP2C19 Genotype-Guided Drug Therapy.

Replace “ persons “ with either patients or subjects, according to need, in all the manuscript

“ Table 1 shows the consumption of PPIs and platelet aggregation inhibitors from 2014 117 to 2018 in Denmark” – could the authors add more recent data?

The word “redeem” is miss-used along the manuscript. Please re-think this.
